# Human and animal skin identified by palaeoproteomics in Scythian leather objects from Ukraine

**Luise Ørsted Brandt**[1], **Meaghan Mackie**[1,2], **Marina Daragan**[3], **Matthew J. Collins**[1,4], **Margarita Gleba**[5] *

1 The Globe Institute, University of Copenhagen, Copenhagen K, Denmark, 2 Novo Nordisk Foundation Center for Protein Research, University of Copenhagen, Copenhagen N, Denmark, 3 Institute of Archaeology of the National Academy of Sciences of Ukraine, Kyiv, Ukraine, 4 University of Cambridge, McDonald Institute for Archaeological Research, Cambridge, United Kingdom, 5 Dipartimento dei Beni Culturali, Università degli Studi di Padova, Padova, Italy

* margarita.gleba@unipd.it

## Abstract

Leather was one of the most important materials of nomadic Scythians, used for clothing, shoes, and quivers, amongst other objects. However, our knowledge regarding the specific animal species used in Scythian leather production remains limited. In this first systematic study, we used palaeoproteomics methods to analyse the species in 45 samples of leather and two fur objects recovered from 18 burials excavated at 14 different Scythian sites in southern Ukraine. Our results demonstrate that Scythians primarily used domesticated species such as sheep, goat, cattle, and horse for the production of leather, while the furs were made of wild animals such as fox, squirrel and feline species. The surprise discovery is the presence of two human skin samples, which for the first time provide direct evidence of the ancient Greek historian Herodotus' claim that Scythians used the skin of their dead enemies to manufacture leather trophy items, such as quiver covers. We argue that leather manufacture is not incompatible with a nomadic lifestyle and that Scythians possessed sophisticated leather production technologies that ensured stable supply of this essential material.

## Introduction

The identification of the animal species used in archaeological leather is important. It not only provides insights into the range of animal species utilised but also reveals choices made in leather production, the functionality and appearance of the leather, and potentially even specific beliefs associated with the objects. Species identification of archaeological skin materials has previously primarily been performed by light and scanning electron microscopy of species-specific traits of the hair follicle pattern of the skin's surface (grain pattern) and, if preserved, the hair morphology, as well as the skin's cross section [1, 2]. Despite being widely applied, the reliability of species identification based on the light and electron microscopic observation of skin and hair morphology is problematic both due to biological variation within species, similarities between species, and loss of diagnostic features due to degradation [3, 4].

**Data Availability Statement:** The data has been made publicly available: https://www.ebi.ac.uk/pride/archive/projects/PXD043533.

**Funding:** This work was supported by the Alexander von Humboldt Stiftung and Gerda Henkel Stiftung (project "The production technology of Scythian Archery equipment: bow, arrows and quivers"; Daragan), and the Danish National Research Foundation (grant 128 PROTEIOS). The funders had no role in study design, data collection and analysis, decision to publish, or preparation of the manuscript.

**Competing interests:** The authors have declared that no competing interests exist.

In recent decades, new methods based on the analysis of ancient biomolecules have been applied for the species identification of hide and leather. The analysis with the highest potential for taxonomic resolution is that of ancient DNA [5–8]. The success of DNA-based approaches, however, depends on DNA preservation, which is conditioned by the diagenetic conditions that the sample experienced during processing (such as tanning) and archaeological deposition [9], both likely to contribute to DNA degradation. More recently, alternative molecular approaches for species identification using mass spectrometry (MS) to analyse proteins have been adopted for the analysis of skin and fur. While liquid chromatography tandem mass spectrometry (LC-MS/MS) based proteomics identifies all proteins in a sample [4], Peptide Mass Fingerprinting (PMF) approaches focus on specific proteins: keratin in fur samples and collagen in skin samples [10, 11]. The latter is also known as Zooarchaeology by Mass Spectrometry (ZooMS). PMF is advantageous for several reasons: first, it requires a minimal sample size; second, it is fast and relatively inexpensive [12], which makes it possible to apply to samples where only minimal samples can be spared as well as to large sample sets; and third, its success has been shown to exceed that of ancient DNA analysis in cases of both older and more degraded samples [4].

We present the results of the first systematic application of PMF to identify species in Scythian leather artefacts excavated in southern Ukraine. The term Scythian is usually applied to a diverse group of nomadic peoples of varied genetic origin who inhabited the vast Eurasian steppes during the first millennium BCE and shared similar material culture, economic structure, lifestyle, and ideology [13–17]. For the purpose of the present study, Scythians are understood to be the nomads that occupied the steppes north of the Black Sea, and between the Danube and the Don Rivers, as defined by the ancient Greek 'father of history' Herodotus [17–19]. For over three centuries (c. 700–300 BCE), Scythians served as the mobile bridge that linked the various sedentary societies of Europe and Asia and played a fundamental role in the creation and transfer of technologies, languages, ideologies, commodities, and pathogens between 'East' and 'West' [15]. Scythian economy and daily life is, however, poorly understood due to their nomadic lifestyle and the disproportionate attention given to the spectacular gold objects from elite graves that have come to characterise the Scythian material culture in the archaeological literature and popular imagination [20–23]. In their everyday life, however, Scythians had much greater need for a stable supply of more basic materials, such as wood, bone, leather, and textile for the production of clothing, tools, and weapons. Among these, leather objects constitute a particularly neglected area of research, as they are usually highly degraded, fragmented, and rather unphotogenic. Degradation issues also result in difficulties in identifying the animal species used to produce these leather items.

Our attempts to use traditional microscopy methods to identify the species of the Scythian leather samples were not successful. Apart from the two fur samples, the rest all lack pelage and their surfaces have been degraded, likely due to scraping and tanning of the skins during their preparation and use, as well as post-depositional processes [24]. Even on the larger fragments, the grain pattern was not sufficiently diagnostic. On the basis of hair morphology, the fur sample from Vil'na Ukraina 3, kurgan 22, burial 1 was identified as belonging to the Mustelidae family, while the fur sample from Ilyinka kurgan 4, burial 3 is likely from a rodent, but the species could not be identified in either case [44].

Since preliminary results of an experimental application of the PMF to some Scythian samples showed that the method was successful [24], we applied ZooMS, using an optimised protocol, to an expanded number of samples: 45 leather samples from 13 different Scythian kurgan sites in Ukraine, primarily dating to the fourth century BCE. One of those samples, as well as two additional samples, were also analysed using LC-MS/MS, bringing the total

number of samples analysed to 47. This was done in order to gain information about the species and a better understanding of leather production in the Pontic Steppe.

## Materials and methods

### Archaeological samples

The archaeological samples used in this study come from 18 burials excavated at 14 different sites in Southern Ukraine (Fig 1). No permits were required for the described study, which complied with all relevant regulations.

Table 1 summarises the contextual information of the archaeological material analysed in this study, including burial type, sex, and age of the deceased (where determined), a burial goods inventory, chronological date, and reference to the report or publication. Osteological analysis was impossible in many cases due to poor bone preservation and the sex in some cases

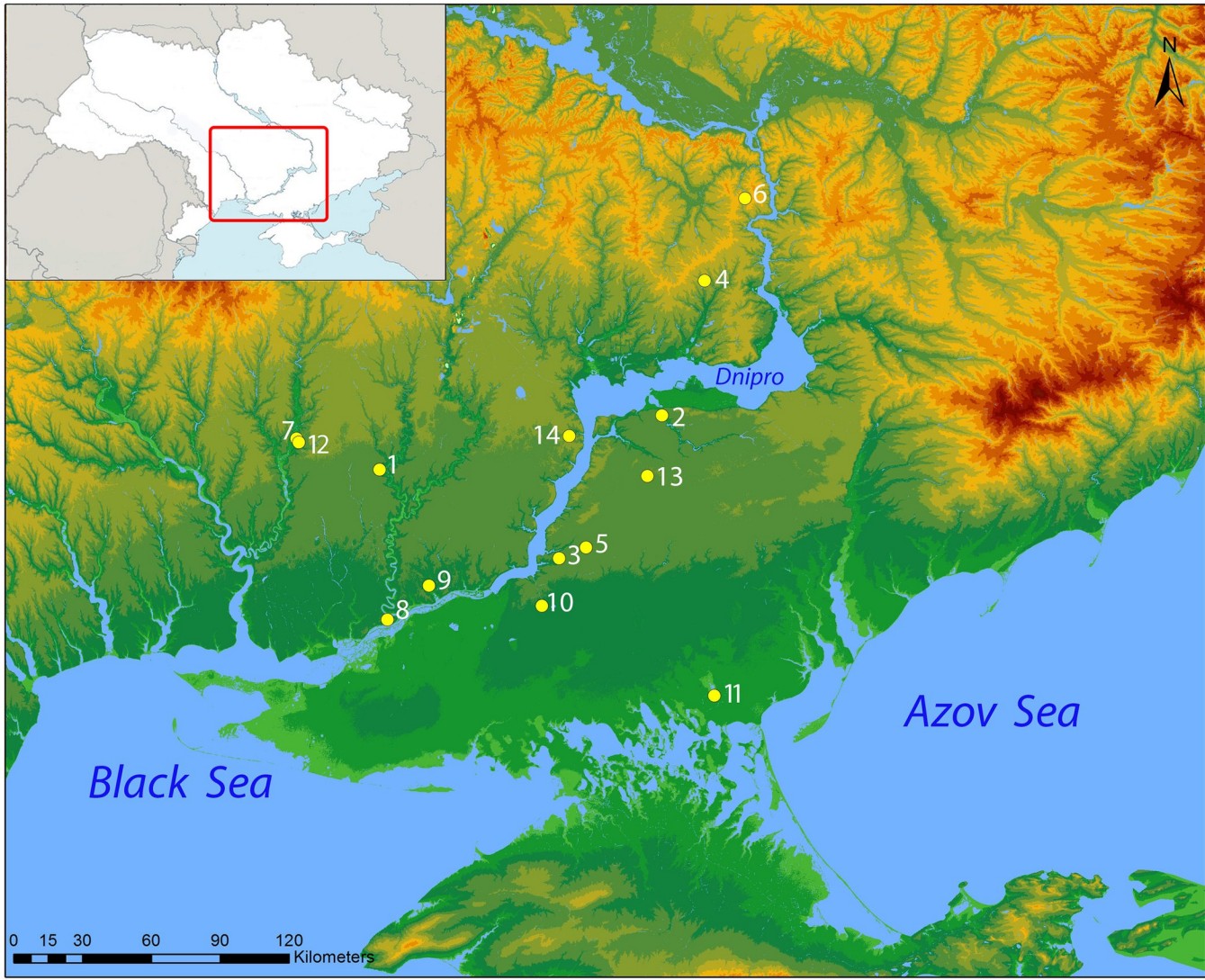

**Fig 1.** The map shows the sites from which leather samples were recovered: 1. Bulhakovo; 2. Ilyinka; 3. Kairy; 4. Kislychevate; 5. Ol'hyne; 6. Orikhove; 7. Otradne; 8. Sadove; 9. Tyahinka; 10. Vil'na Ukraina; 11. Vodoslavka; 12.Vysuns'k; 13. Zelene; 14. Zolota Balka (Map: M. Daragan, based on a DTM STRM Map services and data available from U.S. Geological Survey, National Geospatial Program; Insert: M. Daragan based on https://www.cia.gov/the-world-factbook/countries/ukraine/map).

**Table 1. Sites and context from which archaeological leather items were sampled.** *Without osteological sex determination.

| Site/*Museum* | Burial type | Sex | Burial inventory | Leather remains [sample no.] | Date | Reference | Sample |
|---|---|---|---|---|---|---|---|
| Bulhakovo, kurgan 5, burial 2 Institute of Archaeology of the National Academy of Science of Ukraine, *Kyiv, Ukraine* | Secondary burial. Burial chamber of catacomb type | Double, probably adult male and female* | Wooden boxes; spindles; pottery; Greek pottery; bronze cauldron; iron knives and spears; beads; earrings; grivna; rings; **leather quiver** with arrows | Large fragments of the leather quiver bag and internal partition; preserved *in situ* | Second quarter of the 4th c. BC | [25–27] | 16 (upper back part), 17 (back part), 18 (middle, perforated front part), 21 (decorative ribbon), 22 (front decorated part), 26 (inner partition wall), 30 (lower back part) |
| Ilyinka, kurgan 4, burial 2 Zaporizhzhya Regional Museum, *Zaporizhzhya, Ukraine* | Secondary burial. Burial chamber of catacomb type | Probably adult male | **Nine leather quivers** with arrows; iron spear heads and butts; iron sword in a wooden sheath; iron cuirass and belt; bronze greaves; bronze cauldron; silver kylix; black-figure kylix; golden hrivna; pottery fragment; amphora. | Separate leather quiver fragments (some painted); from three quivers: no. 5, possibly no. 7 and an unknown one | Second-early third quarter of the 4th c. BC | [28, 29] | 7 (quiver 5, indeterminate fragment), 8 (quiver 7, indeterminate fragment), 10 (quiver 5, indeterminate fragment), 11 (indeterminate quiver fragment), 12 (quiver 5, painted part) |
| Ilyinka, kurgan 4, burial 3 Zaporizhzhya Regional Museum, *Zaporizhzhya, Ukraine* | Secondary burial. Burial chamber of catacomb type | Child | **Two leather quivers** with arrows with textile and **fur** remains underneath; iron knives; beads; thin wooden elements, possibly from a bow | Separate leather quiver fragments of the front side (some ornamented and painted) and base; small fur fragments preserving only pellage | Second-early third quarter of the 4th c. BC | [28, 29] | 1 (quiver 2, upper part), 2 (quiver 2, upper part), 3 (quiver 2, middle decorated part), 4 (quiver 2, lower decorated part), 5 (quiver 2, central ribbon), 14 (quiver 5, side decorated part), 40 (quiver, indeterminate part), 48 (fur remains) |
| Ilyinka, kurgan 4, burial 6 Zaporizhzhya Regional Museum, *Zaporizhzhya, Ukraine* | Secondary burial. Burial chamber of catacomb type | Child | Bronze finger rings; beads; black-figure kylix; **leather quiver** with arrows | Separate leather quiver fragments | Second quarter of the 4th c. BC | [28, 29] | 9 (indeterminate quiver fragment) |
| Kairy, kurgan group V, kurgan 1, burial 1 Institute of Archaeology of the National Academy of Science of Ukraine, *Kyiv, Ukraine* | Main burial. Oval pit burial | Male, 25–30 yo | Wooden dishes; iron spears and finial; bronze greaves; **leather quiver** with arrows; **leather trousers** | Quiver base and unidentified fragments of the quiver bag; and possible fragment of trousers | End of the 5th-first quarter of the 4th century BC | [30] | 23 (quiver bottom), 25 (quiver side or trousers), 47 (indeterminate quiver part) |
| Kyslychevate, kurgan group 2, kurgan 7, burial 1 Museum of Archaeology of Oles Honchar, Dnipro National University, *Dnipro, Ukraine* | Main burial; burial chamber of catacomb type | Unknown | Coral pendants; iron awls; lead weight; arrowhead; embossed **leather fragments** | Possible shoe fragments | 4th century BC | [31] | 39 (indeterminate embossed fragment) |
| O'hyne, kurgan 2, burial 1 Institute of Archaeology of the National Academy of Science of Ukraine, *Kyiv, Ukraine* | Main burial. Burial chamber of catacomb type | Looted (a few bones survive) | Arrowheads; **leather quiver fragments**; kylix; bronze finial; iron knife and spear butts | Quiver base and unidentified fragments of the quiver bag | 2nd quarter of the 4th c. BC | [32, 33] | 24 (quiver bottom) |

*(Continued)*

**Table 1.** (Continued)

| Site/*Museum* | Burial type | Sex | Burial inventory | Leather remains [sample no.] | Date | Reference | Sample |
|---|---|---|---|---|---|---|---|
| Orikhove, kurgan 3 burial 2 Museum of Archaeology of Oles Honchar, Dnipro National University, *Dnipro*, *Ukraine* | Secondary burial. Burial chamber of catacomb type | Double male* | **Three leather quivers** with arrows; iron spears, belt and knife | Quiver base and separate fragments of the quiver bag, some ornamented | First half of the 4th c. BC | [34] | 30a (quiver 3, quiver bottom), 31 (quiver 1, side part), 32 (quiver 2, indeterminate part), 33 (quiver 2, decorative part), 34 (quiver 2, decorative part with stitching), 35 (quiver 2, indeterminate part) |
| Otradne kurgan 3, burial 1 Institute of Archaeology of the National Academy of Science of Ukraine, *Kyiv*, *Ukraine* | Burial chamber of catacomb type. | Probably male* | Iron knife, spear head, dart, bone finial, **leather quiver** with arrows | Unidentified fragments of leather quiver or boot | Second quarter of the 4th c. BC | [35] | 43 (indeterminate fragment) |
| Otradne kurgan 3, burial 2 Institute of Archaeology of the National Academy of Science of Ukraine, *Kyiv*, *Ukraine* | Burial chamber of catacomb type | Probably female* | Plate, Greek kantharos, spindle, lead spindle whorls, iron spits, Greek transport amphora, silver finger rings and pendant, earrings, bronze mirror, pectoral, glass beads, iron knife, spearheads, darts, arrow heads, **leather fragment** | Unidentified fragments of leather, possibly quiver | Second-third quarter of the 4th c. BC | [35] | 42 (indeterminate fragment) |
| Sadove, kurgan 4, burial 1 Institute of Archaeology of the National Academy of Science of Ukraine, *Kyiv*, *Ukraine* | Secondary burial. Burial pit | Adult* | Knife; **leather quiver** with arrows | Indeterminate fragments of the quiver bag | Third quarter of the 5th century BC | [32, 36] | 15 (indeterminate quiver fragment) |
| Tyaginka kurgan 8, burial 3 Institute of Archaeology of the National Academy of Science of Ukraine, *Kyiv*, *Ukraine* | Secondary burial; burial chamber of catacomb type with two entrance pits | Unknown | Bronze cauldron, amphorae, iron knives, kylix, finial, sword, golden pendants, beads, bronze mirror, iron spearheads, dart, bronze 'fork', iron horse bit and psialia, **two leather quivers** with arrows | indeterminate fragments of quiver | Mid-4[th] c. BC | [37, 38] | 44 (indeterminate quiver fragment) |
| Vil'na Ukraina 3 kurgan 22 burial 1 Institute of Archaeology of the National Academy of Science of Ukraine, *Kyiv*, *Ukraine* | Secondary burial: burial chamber of catacomb type | Adult female | Bronze mirror with iron handle and textile and **fur** remains; golden fittings of a mirror case; golden elements of headdress; earrings; applique; black-figure bowl; two bronze spindle whorls; wooden trays with sacrificial food remains; iron knives with bone handles; bronze plate; glass beads | A layer of fur pellage | Second to third quarter of the 4th century BC | [39] | 49 (fur remains) |

*(Continued)*

**Table 1.** (Continued)

| Site/*Museum* | Burial type | Sex | Burial inventory | Leather remains [sample no.] | Date | Reference | Sample |
|---|---|---|---|---|---|---|---|
| Vodoslavka, kurgan 8, burial 4 Institute of Archaeology of the National Academy of Science of Ukraine, *Kyiv, Ukraine* | Secondary burial. Burial chamber of catacomb type | Sub-adult, probably male* | **Leather quiver** with arrows; bow; bronze finial; iron knives; wooden plate | Separate leather quiver fragments of the front side with one ornamented piece | Beginning of the third quarter of the 4th c. BC | [40, 41] | 19 (front decorated quiver part), 27 (external quiver part), 28 (indeterminate quiver part), 41 (indeterminate quiver part), 46 (indeterminate quiver part) |
| Vysuns'k kurgan 6, burial 1 Institute of Archaeology of the National Academy of Science of Ukraine, *Kyiv, Ukraine* | Burial chamber of catacomb type with two entrance pits | Two skeletons* | Iron knives, Greek kantharos, bronze mirror with **leather cas**e, bracelets, glass beads, stone, arrow heads, leather quiver with arrows | Separate fragments of leather mirror case | Second quarter of the 4th c. BC | [37, 42] | 45 (mirror case fragment) |
| Zelene, kurgan group I, kurgan 2, burial 3 Kherson Regional Museum, *Kherson, Ukraine* | Secondary burial. Burial chamber of catacomb type with two chambers | Chamber 1 – male burial; chamber 2—female burial | Iron cuirass, hip guards; spear heads and butts, knife; wooden dishes; stool; stone; lekythos; kylix; **two leather quivers**, **leather shoes** | Separate leather fragments of the quiver bag and possible shoe fragment | End of the 5th-early 4th century BC | [43] | 36 (chamber 2, indeterminate part), 37 (chamber 1, quiver 1, indeterminate part), 38 (chamber 2, indeterminate part) |
| Zolota Balka, kurgan 13, burial 7 Institute of Archaeology of the National Academy of Science of Ukraine, *Kyiv, Ukraine* | Secondary burial. Burial chamber of catacomb type | Male | Golden earrings and finger ring; two leather quivers with arrows; iron spear heads, darts, knife; bronze finial; bone and wooden objects; **leather vessel**; amphora | Top part of a leather vessel | Beginning of the third quarter of the 4th c. BC | [32] | 20 (top of leather vessel) |

is therefore ascribed based on the burial inventory. Not all originally recovered pieces have survived. Detailed descriptions and images of samples can be found in Table S1 (S1 File). The location of the objects at the time of sampling is noted in the first column.

In many cases, the function of the original leather object cannot be determined with certainty due to their extremely fragmentary state of preservation (Fig 2). Some are likely elements of clothing (possibly trousers), boots or vessels. The majority of the leather artefacts, however, constitute the remains of some of the most iconic Scythian objects: quivers, containers for arrow storage, or *gorytoi*, containers that provided a secure storage and transportation of both arrows and the bow (Fig 3). Scythian archers are invariably depicted carrying a quiver/ *gorytos* in ancient iconography and almost every Scythian burial is accompanied by a quiver set, although usually only the metal arrowheads survive [27]. Quivers are often depicted in artistic and decorative objects of the time. For example, a *gorytos* hangs above the central scene of the upper frieze of the gold pectoral from Tovsta Mohyla (Fig 4). The best preserved quivers from Bulhakovo and Ilyinka have allowed a reconstruction of the construction methods [27], but little was known about the nature of the materials used for the manufacture of the more mundane examples of leather Scythian quivers until the present study.

## ZooMS analysis

A total of 45 archaeological leather items were sampled for ZooMS analysis. From each item a sample of a minimum of 2x2mm was discretely removed for analysis. The samples were

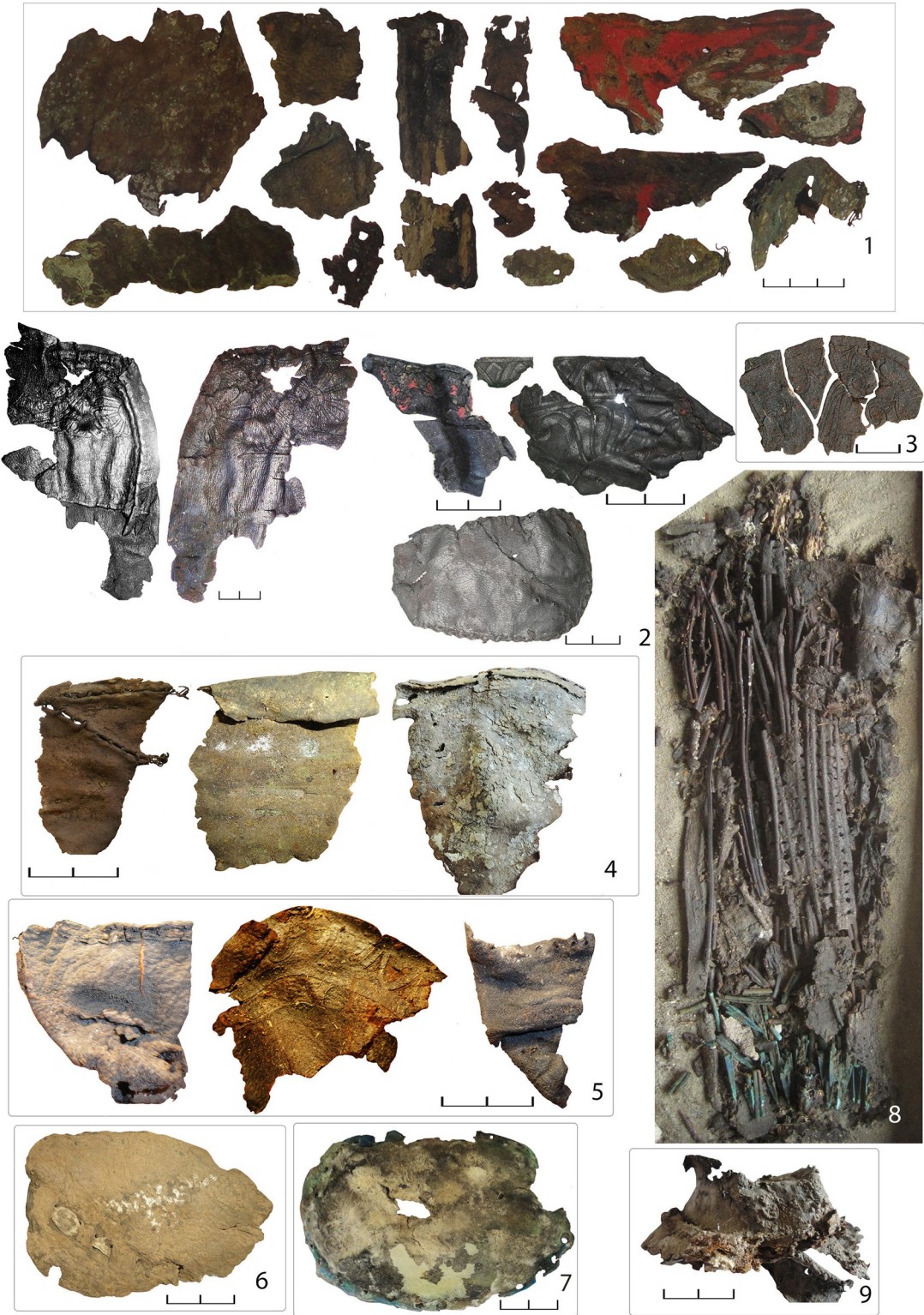

**Fig 2.** A selection of the leather object fragments analysed in this study: 1. Ilyinka kurgan 4 burial 2; 2. Ilyinka kurgan 4 burial 3; 3. Vodoslavka kurgan 8 burial 4; 4. Orikhove kurgan 3 burial 2; 5. Zelene I kurgan 2 burial 3; 6. Kairy V kurgan 1 burial 1; 7. Ol'hyne kurgan 2 burial 1; 8. Bulhakovo kurgan 5 burial 2; 9. Zolota Balka kurgan 13 burial 7 (Image: M. Daragan). The units of the scale bars are cm.

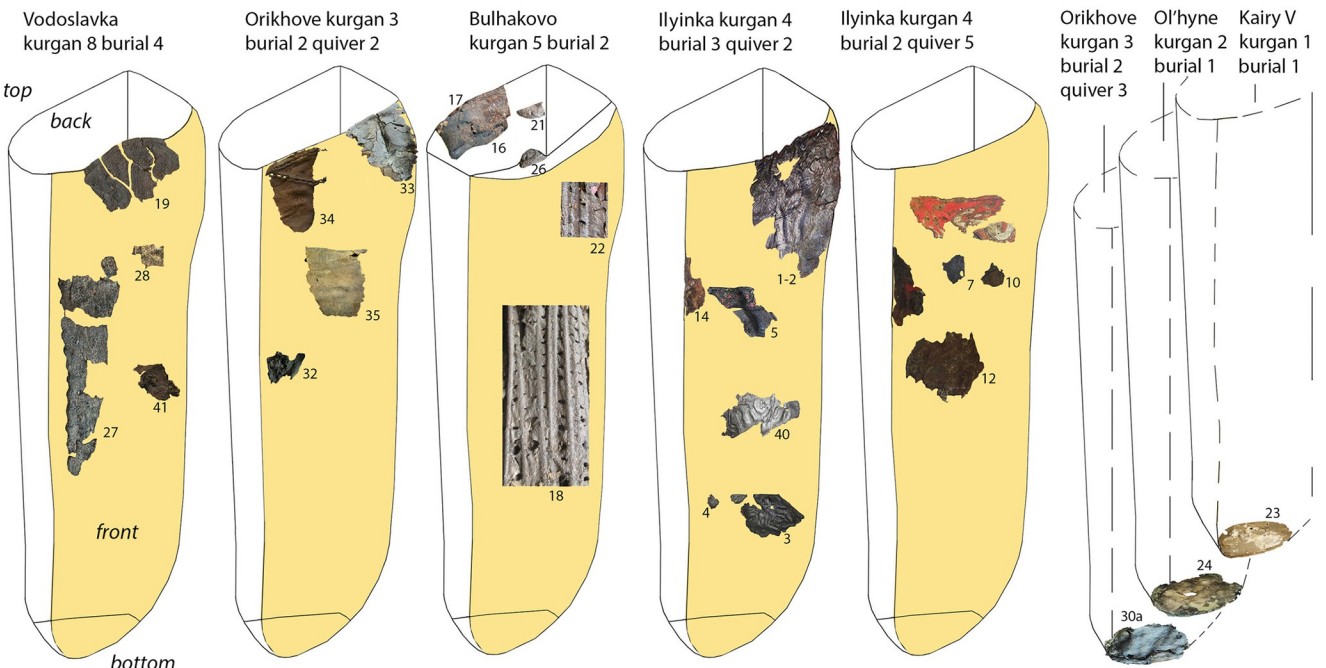

**Fig 3. Graphic reconstruction of quivers with approximate location of their extant fragments; numbers indicate sample numbers (Image: M. Daragan).**

prepared according to a previously published protocol for parchment [45] with previously described optimisations for archaeological leather [3], this time including a gelatinisation of one hour at 65˚C before the trypsin digestion.

Samples were prepared at University of Copenhagen in the laboratory facilities of the Globe Institute. After spotting, plates were sent for MALDI ToF-analysis at BioArCh, University of York. Mass spectra were acquired over the m/z range 800–4000. Spectral analysis was performed using the open-source software mMass (www.mmass.org) [46]. The three spectra generated for each sample were averaged, and the average spectrum was then inspected manually for the presence of previously described peptide markers which may vary between species [47–49]. Taxonomic identifications were assigned at the most conservative level of identification (genus or family) based on the presence of unambiguous markers. These are marked with '?' if present at low intensity, or with a low signal to noise (S/N) ratio. Spectra of poor quality (i.e. low S/N ratio and with no distinct markers) were designated 'No ID'. The reference database of animals for which peptide markers have been identified is constantly growing. Currently the database contains all common domesticated mammals, and numerous species within the biological families Cervidae, Bovidae, Equidae, Canidae, Felidae, Mustelidae, and Hominidae, which would be relevant to the material in question.

### Liquid chromatography-tandem mass spectrometry (LC-MS/MS)

One sample, 21, which showed low resolution albeit a potentially important result, was chosen for re-analysis by LC-MS/MS.

Two samples (48, 49) for which only fur and not skin was preserved were initially analysed by peptide mass fingerprinting using the protocol presented by Brandt et al. [50] with poor results. As mentioned above, hair morphology indicated Rodentia and Musteliadae, respectively, as the potential sources. These were therefore also chosen for LC-MS/MS.

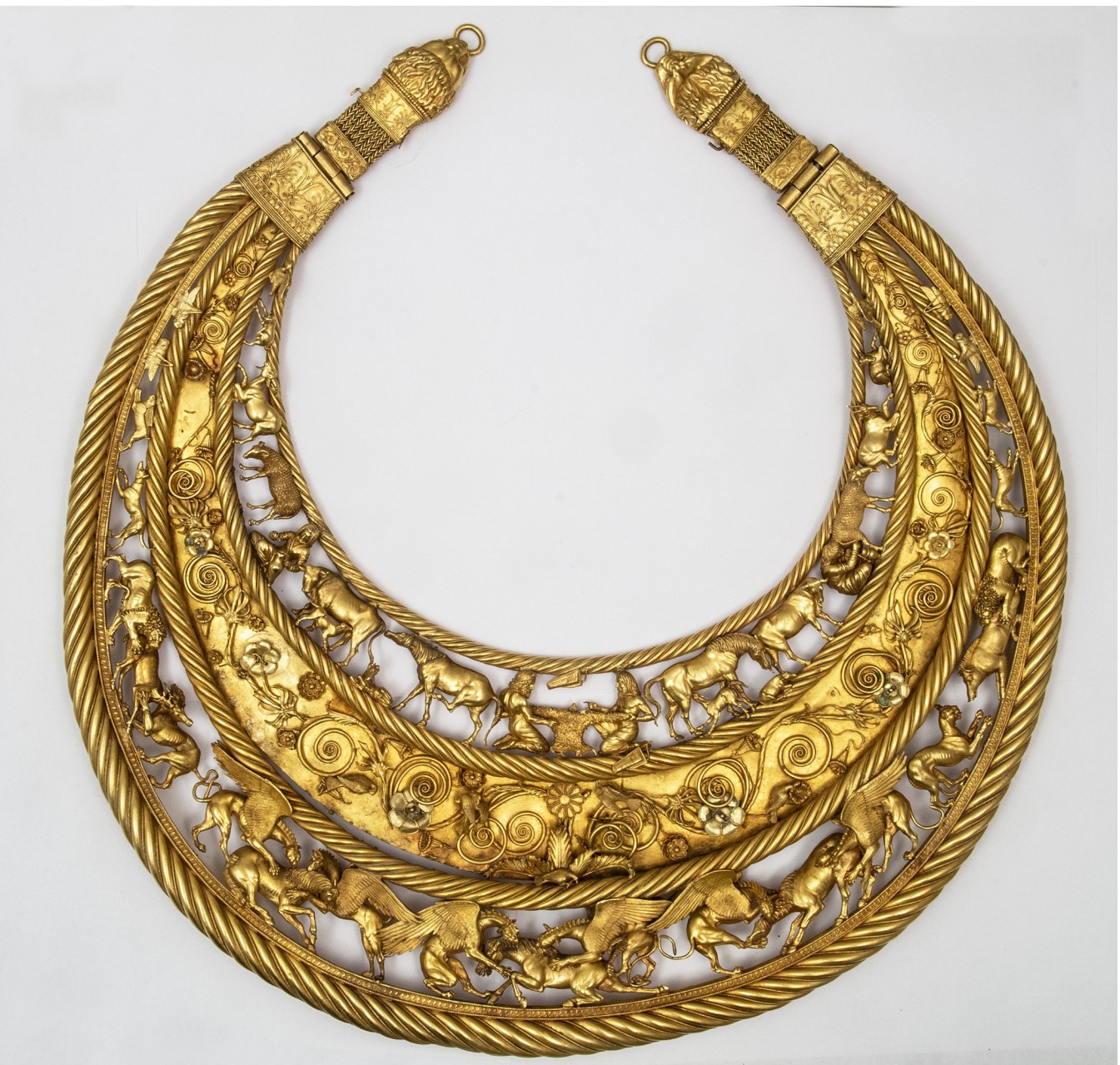

**Fig 4. Scythian gold pectoral from Tovsta Mohyla, Ukraine (Image: After Polidovich 2021).**

For LC-MS/MS analysis, ZooMS eluates were provided at approximately 0.4 μg/μL in 50% acetonitrile (ACN). Of this, 3 μL of each sample was put into separate wells of a 96-well MS plate. Samples were then vacuum-centrifuged to remove the ACN. Samples were then resuspended in 12 μL of 5% ACN 0.1% trifluoroacetic acid (TFA) and analysed by LC-MS/MS based on previously published protocols for palaeoproteomic samples [50, 51].

For the data analysis, Thermo RAW files generated by the Exploris 480 mass spectrometer (Thermo Scientific, Bremen, Germany) were then searched using MaxQuant (v.1.6.3.4; [52]) for interpretation of spectra. Files were first run against the Swissprot Database (downloaded from Uniprot 09/06/2020) and the common contaminant database provided by MaxQuant (of

which identifications were removed from analysis). From this initial search, iterative searches were made to narrow down the database to the relevant proteins from the relevant species. Detailed LC-MS/MS and data analysis methods as well as spectra are provided in the S1 and S2 Files.

## Results

### Species identification

Thirty-three of 45 samples analysed with ZooMS were identified, which is within the expected success rate for archaeological leather [53, 54]. Of these, 16 were identifications to a single species, 4 were probable identifications to a single species, 7 to family level, 1 to a probable family, and 5 were identified to one of two or more species (Table 2). The majority of the ZooMS-identified species are domesticates (Fig 5), with more than half of the identified species being either goat or sheep. Based on the marker 3093, sample 33 from Orikhove is identified to *Capra hircus/Rangifer tarandus* rather than sheep (as determined by Spindler et al. [24]). There is also one example of cattle (sample 12), and a further three from either the Bovidae or Cervidae families, although due to poor collagen preservation we were not able to make a more distinct identification. Two samples come from horse (11 and 14).

The final three ZooMS identifications stand out the most: two come from a wild carnivore (one identified as red fox (sample 45), the other could be referred to either tiger, lion, marten, wolverine, otter or hyena (sample 34). The final sample (7), from a quiver found in Ilyinka, kurgan 4, burial 2, is of particular interest as it is identified as human (Fig 6). Observed peptide markers can be found in S1 Dataset.

The three samples investigated by LC-MS/MS can conservatively be identified as Sciuridae (Sample 48), Felidae (Sample 49) and Homininae (Sample 21) (Table 3). Additional tables on proteins present and peptide specificity, as well as spectra of peptides relevant for species identification, can be found in the SI.

## Discussion

Leather from different animal species requires different preparation methods. Our results, and the range of species identified, demonstrate that Scythians possessed a sophisticated knowledge of skin processing, indicating a certain degree of craft specialisation. The results also indicate that leather was possibly chosen based on physical properties, availability, or potentially for cultural reasons.

### Domestic animal exploitation

The majority of the samples come from domesticated species which would be herded on the steppe, and represented the major part of the economy and wealth of the pastoralist nomads. Sheep and goat are the most abundantly identified species, and therefore are assumed to be preferred sources of skin. This may be explained by either the specific properties of their skin or their availability. Goat skin is very soft but also extremely durable, flexible, and water resistant, while sheep skin is soft, smooth and light, making them very suitable materials for shoes and quivers. Cattle skin, on the other hand, is thick and durable [56].

Horsehide is a particularly tough, durable and resistant type of leather. Horse skin with pellage was identified in the outer garments from burials excavated at Suglug-Kem I, Sypuchiy Yar, Sausken 3, Eki-Otdug I, and Bay-Dag 5 burials in the central Tuva region of Russia [57].

All of these materials were readily available as goats, sheep, cattle, and horses were the principal domestic animals kept and bred by the pastoralist Scythians, as depicted on the exquisite

**Table 2. ZooMS identifications of the 45 samples.** *This ID cannot be distinguished from *Pan troglodytes* (chimpanzee) and *Pan paniscus* (bonobo), who are however both African species and would be very surprising in this cultural context. **Only a few species within the genus *Mustela* are currently included in the ZooMS database and these cannot be distinguished [49, 55].

| Site | Burial | Sample ID | ZooMS ID, Latin name | ZooMS ID, common name |
|---|---|---|---|---|
| Bulhakovo | Kurgan 5, burial 2 | 16 | No ID | |
| | | 17 | No ID | |
| | | 18 | *Capra hircus* | Dometicated goat |
| | | 21 | Primates? | Primate? |
| | | 22 | No ID | |
| | | 26 | *Capra hircus*? | Domesticated goat? |
| | | 30 | *Capra hircus* | Domesticated goat |
| Ilyinka | Kurgan 4, burial 2 | 7 | *Homo sapiens** | Human |
| | | 8 | *Capra hircus* | Domesticated goat |
| | | 10 | *Capra hircus* | Domesticated goat |
| | | 11 | Equidae | Horse Family |
| | | 12 | *Bos taurus* | Domesticated cattle |
| | Kurgan 4, burial 3 | 1 | *Ovis aries* | Domesticated sheep |
| | | 2 | *Ovis aries*? | Domesticated sheep |
| | | 3 | No ID | |
| | | 4 | No ID | |
| | | 5 | *Capra hircus* | Domesticated goat |
| | | 14 | Equidae | Horse Family |
| | | 40 | *Ovis aries* | Domesticated sheep |
| | Kurgan 4, burial 6 | 9 | No ID | |
| Kairy | Kurgan group V, kurgan 1, burial 1 | 23 | *Capra hircus*/*Rangifer tarandus* | Domesticated goat/reindeer |
| | | 25 | *Capra hircus* | Domesticated goat |
| | | 47 | *Capra hircus*/*Rangifer tarandus* | Domesticated goat/reindeer |
| Kislychevate | Kurgan group 2, kurgan 7, burial 1 | 39 | No ID | |
| Ol'hyne | Kurgan 2, burial 1 | 24 | Bovidae/cervidae | The family of bovids or cervids, cloven-hoofed or hoofed, ruminant mammals |
| Orikhove | Kurgan 3, burial 2 | 30a | *Ovis aries* | Domesticated sheep |
| | | 31 | Bovidae/cervidae | The family of bovids or cervids, cloven-hoofed or hoofed, ruminant mammals |
| | | 32 | No ID | |
| | | 33 | *Capra hircus*/*Rangifer tarandus* | Domesticated goat/reindeer |
| | | 34 | Carnivora** | Carnivore |
| | | 35 | No ID | |
| Otradne | Kurgan 3, burial 1 | 43 | *Capra hircus* | Domesticated goat |
| | Kurgan 3, burial 2 | 42 | *Ovis aries*? | Domesticated sheep? |
| Sadove | Kurgan 4, burial 1 | 15 | Bovidae | The family of bovids, cloven-hoofed, ruminant mammals |
| Tyahinka | Kurgan 8, burial 3 | 44 | Bovidae/cervidae | The family of bovids or cervids, cloven-hoofed or hoofed, ruminant mammals |
| Vysuns'k | Kurgan 6, burial 1 | 45 | *Vulpes vulpes* | Red fox |

(*Continued*)

**Table 2.** (Continued)

| Site | Burial | Sample ID | ZooMS ID, Latin name | ZooMS ID, common name |
|---|---|---|---|---|
| Vodoslavka | Kurgan 8, burial 4 | 19 | *Capra hircus/Rangifer tarandus* | Domesticated goat/reindeer |
| | | 27 | No ID | |
| | | 28 | Bovidae/cervidae | The family of bovids or cervids, cloven-hoofed or hoofed, ruminant mammals |
| | | 41 | *Ovis aries* | Domesticated sheep |
| | | 46 | No ID | |
| Zelene | Kurgan group I, kurgan 2, burial 3 | 36 | *Ovis aries* | Domesticated sheep |
| | | 37 | *Ovis aries* | Domesticated sheep |
| | | 38 | *Ovis aries* | Domesticated sheep |
| Zolota Balka | Kurgan 13, burial 7 | 20 | No ID | |

gold pectoral from Tovsta Mohyla [15, 58] (See Fig 3). Additionally, goat and sheep bones have been found as funeral offerings within the kurgans and among the funeral feast remains from Alexandropol and other Scythian kurgans, as well as in settlements [59]. Quivers have a complex structure with multiple elements, each of which could have required different properties. Therefore, it would not be unusual to use different types of leather to fit different purposes, as observed in the quivers from Bulhakovo and Vodoslavka. At the same time, the use of diverse species within the same quiver may also suggest that leather which was most readily available at the moment may have been used for their construction.

## Wild animal exploitation

A minor proportion of the samples derive from wild animals hunted for their fur: a red fox, a type of cat, and a member of the squirrel family. None of these have been identified in the Ukrainian Scythian zooarchaeological record. It is likely that more hunted animals would be identified if more clothing samples had been analysed, where the fur would be used for display, as compared to the dehaired domesticate leather of the quivers. Indeed, the Scythian archaeological furs found in Ukraine and across Eurasia represent parts used in clothing, and in their majority derive from wild animals of rodent and carnivore families, such as hare, squirrel, jerboa, mole, weasel, stout, ermine, sable [57, 60, 61].

## Human skin

The ancient Greek 'father of history' Herodotus dedicated an entire book to describing Scythian history and customs, and reported upon some remarkable stories. Among these, are tales of Scythians drinking blood of their enemies, using human scalps as trophies and flaying their dead enemies in order to turn the skin into a leather cover for their quivers:

'A Scythian drinks the blood of the first man whom he has taken down. He carries the heads of all whom he has slain in the battle to his king; for if he brings a head, he receives a share of the booty taken, but not otherwise. He scalps the head by making a cut around it by the ears, then grasping the scalp and shaking the head off. Then he scrapes out the flesh with the rib of a steer, and kneads the skin with his hands, and having made it supple he keeps it for a hand towel, fastening it to the bridle of the horse which he himself rides, and taking pride in it; for he who has most scalps for hand towels is judged the best man. Many Scythians even make garments to wear out of these scalps, sewing them together like coats

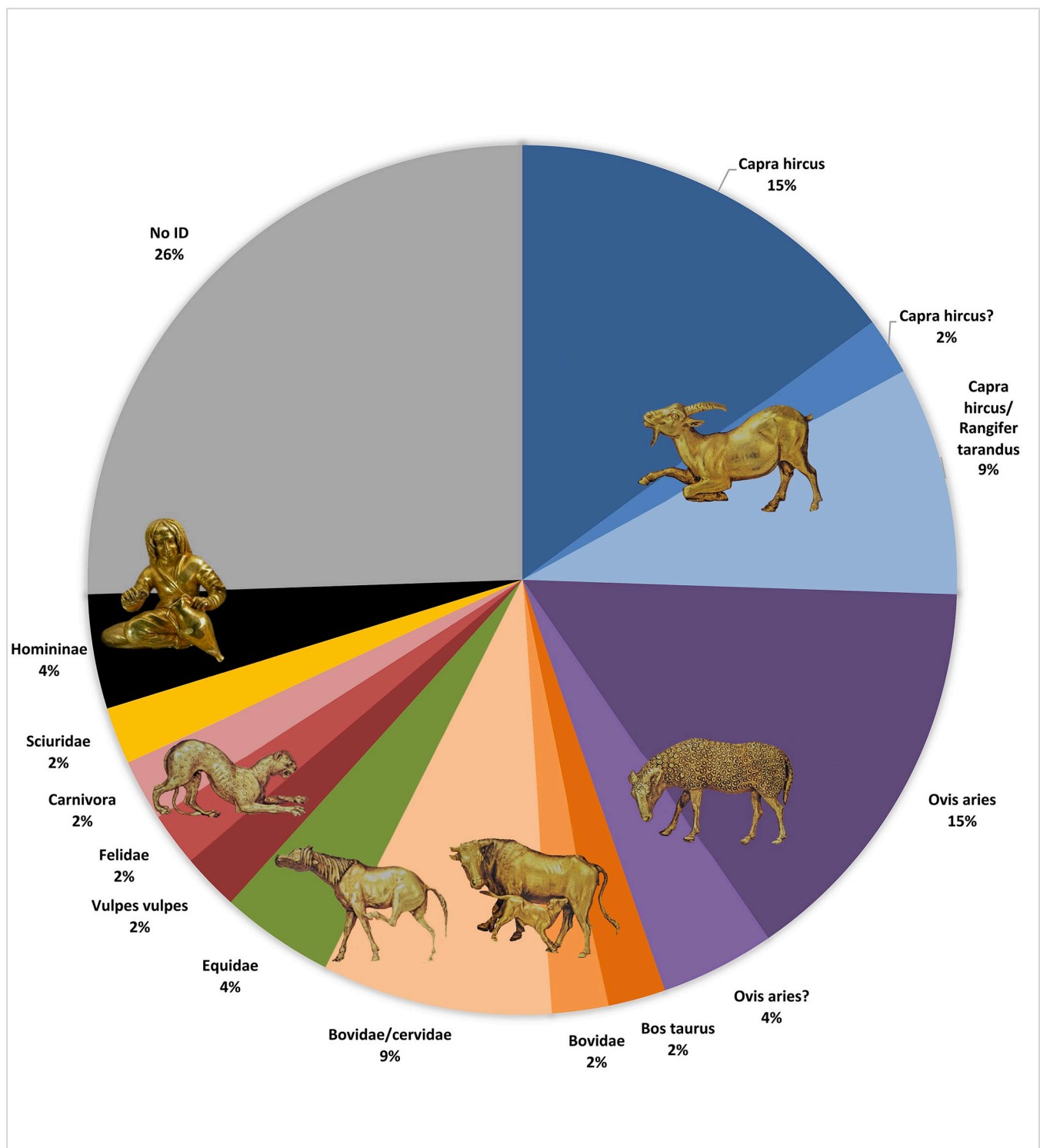

**Fig 5. Pie chart of identifications from ZooMS and LC-MS/MS analysis (Image: Authors).**

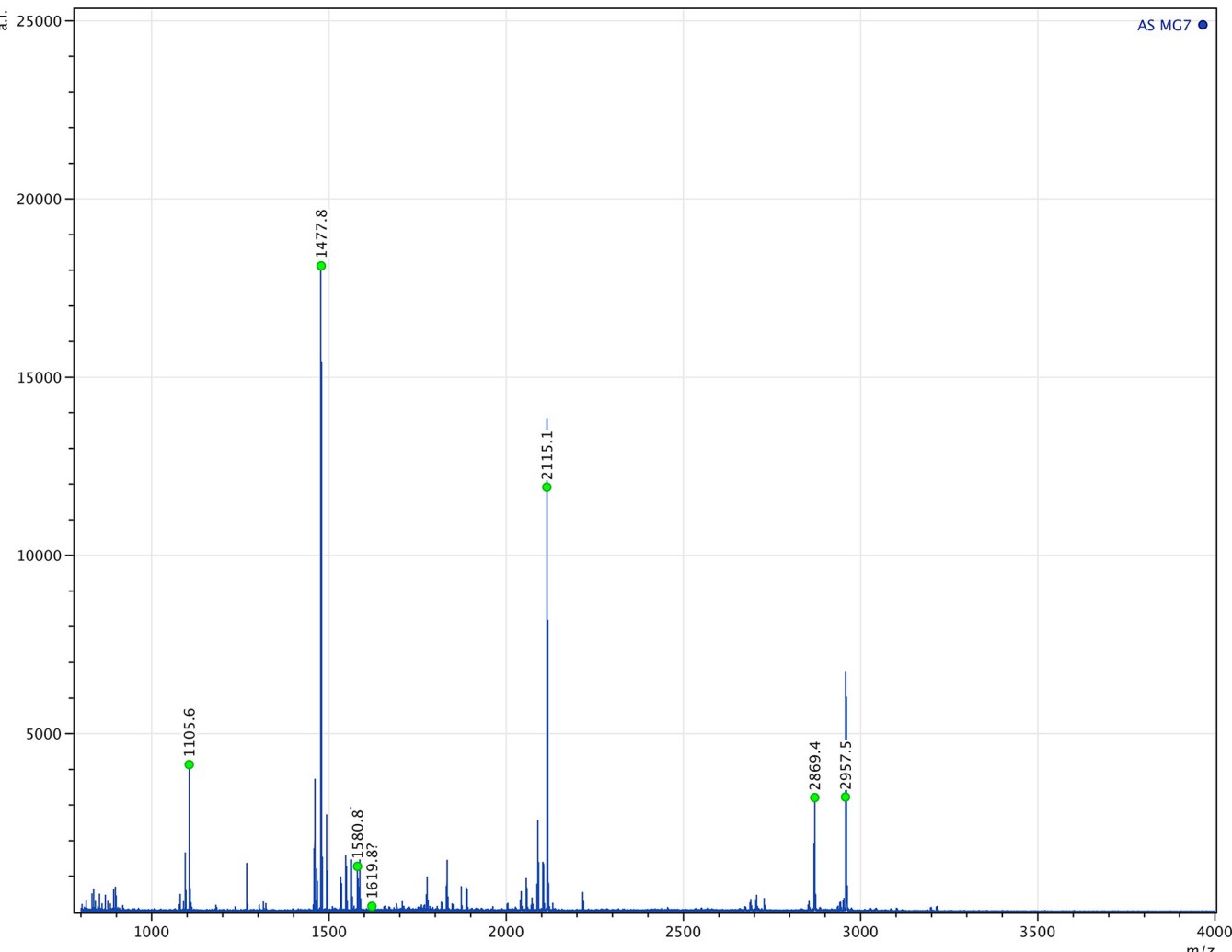

**Fig 6. Spectrum of sample 7 from Ilyinka kurgan 4 burial 2 with markers identifying it to human skin.**

of skin. Many too take off the skin, nails and all, from their dead enemies' right hands, and make coverings for their quivers; the human skin was, as it turned out, thick and shining, the brightest and whitest skin of all, one might say. Many flay the skin from the whole body, too, and carry it about on horseback stretched on a wooden frame.' (Herodotus, Histories, 4.64).

Our results appear to confirm Herodotus' grizzly claim. The proteins from samples 7 and 21, deriving from quivers found in Ilyinka kurgan 4 burial 2 and Bulhakovo kurgan 5 burial 2, indicate human origin, based on both ZooMS and LC-MS/MS analysis. To our knowledge, the

**Table 3. Number of proteins and corresponding peptides identified in each sample.** No. of MS2 spectra is the total for identified proteins. *This is based on current databases and potential *S. vulgaris* keratin sequence fragments [50] ** No single match in current databases suggests it is a Eurasian felid; currently not sequenced.

| Site | Burial | Sample ID | Species ID | No. of Proteins | No. of Identified Peptides | No. of MS2 spectra |
|---|---|---|---|---|---|---|
| Bulhakovo | Kurgan 5, burial 2 | **21** | Homininae (probably *Homo sapiens*) | 37 | 337 | 444 |
| Ilyinka | Kurgan 4 burial 3 | **48** | Sciuridae, most likely *Sciurus vulgaris** | 10 | 279 | 1238 |
| Vil'na Ukraina 3 | Kurgan 22 burial 1 | **49** | Felidae** | 13 | 992 | 2140 |

only other reported instance of the use of human skin in a quiver was identified in Yakovlevskiy kurgan 3 burial 1 (4th century BC) in Russia using counter-immunoelectrophoresis [62].

Although macabre to our modern view, other Scythian customs described by Herodotus have also been supported by archaeological findings [63]. For example, the recent re-investigation of one of the four largest royal Scythian kurgans in southern Ukraine, the Aleksandropol mound, led to the discovery of a large funerary feasting area in the immediate vicinity of the kurgan and, within it, 11 accompanying burials of men, women and children, all of whom appear to have been killed and buried there as part of the funerary rites for the royal occupant of the burial mound [64]. These details closely correspond to Herodotus' description of a Scythian king's funeral (Herodotus 4.71–72; [65]). The description of how mourners would carry out self-mutilation during burials of kings to express their grief has also been confirmed by the excavation of the burial mound of Chortomlyk. Here, six phalanxes of human fingers, two with cut marks, belonging to three or four different people were found, suggesting that Scythians did in fact mourn their kings by cutting off fingers [66].

## Leather uses and functions

In the case of quivers where multiple samples were analysed (Fig 3), it was discovered that the leather of multiple species was used in their construction. Thus, the quiver from Vodoslavka combined goat, sheep and unidentified bovid species; the quiver from Bulhakovo had one of the top elements in human leather and others in goat skin; Orihove quiver combined the top element in carnivore skin with goat, sheep and bovid skin; Ilyinka burial 2 quiver had human, goat, horse and cattle skin, while the quiver from burial 3 had sheep, goat and horse. The fact there appears to be no apparent pattern in species usage for different quiver elements suggests that there was no strict standardisation of raw materials for quiver production. The more unusual, human and carnivore, leather appears to have been used in the top parts of the quivers. This may indicate that each archer made their own quiver using the materials available at the moment.

The two fur elements from Vil'na Ukraina and Vysuns'k associated with bronze mirrors, and the fur from Ilyinka are wild animal species: some type of cat, red fox and a member of the squirrel family. Their use was thus reserved.

## Economy of Scythian leather production

Scythian archaeological and iconographic evidence provides proof that leather was used to make vessels, mirror cases, quivers [27], shoes [26, 67], garments such as trousers and coats [68], and the lining for metal armour such as greaves [69]. The detailed iconographic depictions of Scythians in leather garments with embossed decoration such as those appearing on Scythian toreutics [15, 68] (Fig 7), appear to show distinctly Scythian fashion. Leather was also an important structural element of scale armour, since the metal scales were sewn onto a leather base. Leather can thus be seen as an essential material among the Scythian populations. The results of our analyses provide important new information regarding the production and use of leather by the Pontic Scythians. Ethnographic evidence indicates that the nomads of Kazakhstan and Eastern Tibet made leather products themselves [70, 71]. With regard to the Scythians, we may assume the same. Indeed, the central scene of the spectacular Tovsta Mohyla pectoral appears to show two Scythian men engaged in either skinning a sheep or production of a garment made of sheepskin [72] (Fig 4).

Nonetheless, it should be noted that the embossed decoration on some the studied quiver fragments, for example those from Ilyinka, Vodoslavka, Zelene (Fig 2; 2, 3, 5), as well as some from Alexandropol [73], have a distinctly Hellenic style. Since the Scythians of southern

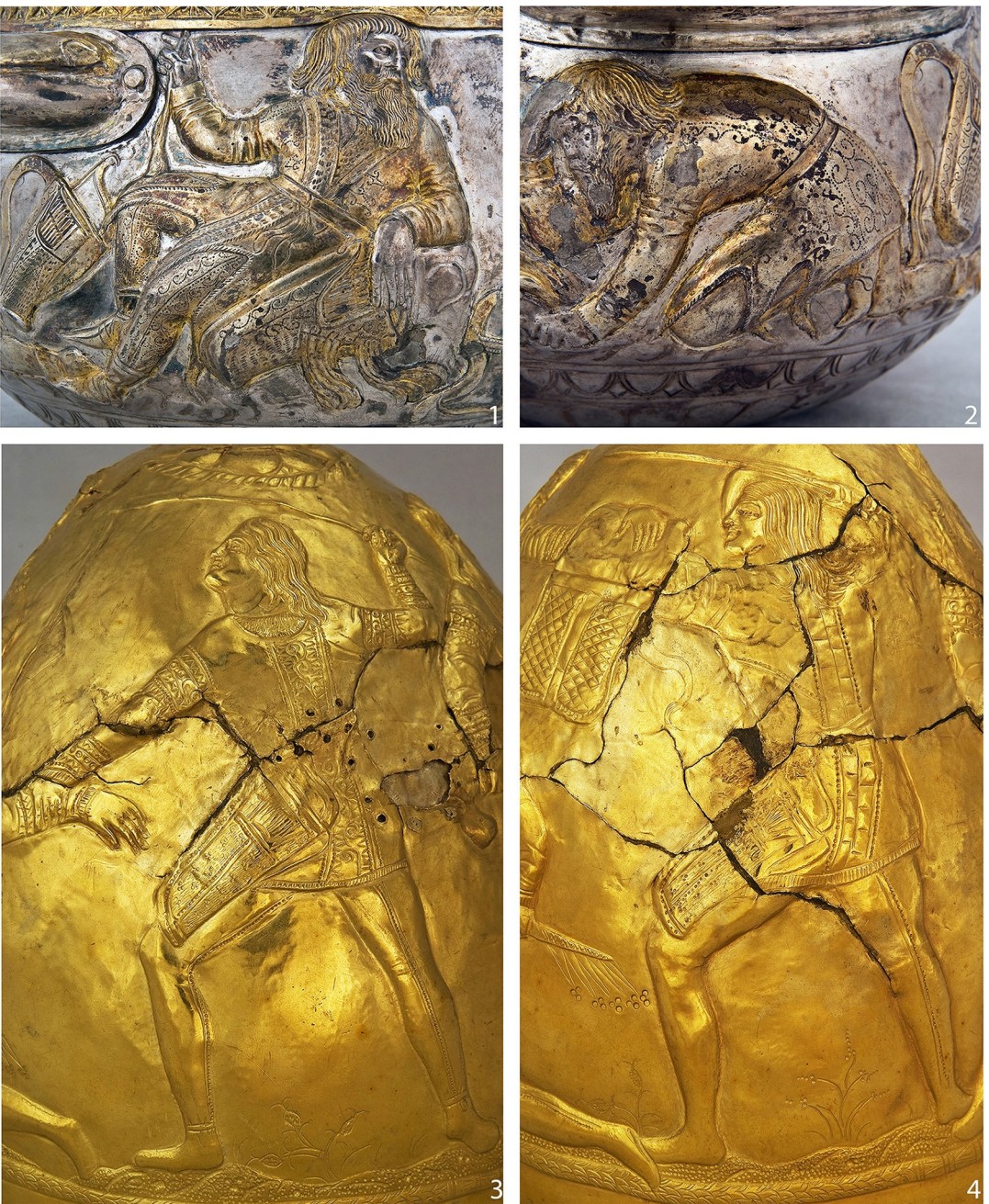

**Fig 7.** Depictions of Scythian warriors wearing decorated sleeved leather garments: 1–2. Guilded silver bowl from Haimanova Mohyla, north chamber (After Bidzilia, Polin 2012, Figs 150, 158); 3–4. Golden cone from Perederiyiva Mohyla, Ukraine (Image: Museum of Historical Treasures of Ukraine, Kyiv).

Ukraine obtained a lot of materials and objects through exchange with the Pontic Greeks, there is the possibility that some of the quivers were produced or at least decorated in Greek settlements on the northern coast of the Black Sea. An alternative explanation would see Scythian craftspeople producing objects decorated in a Greek style. This is hardly unexpected, given that the scabbards of Scythian swords and many other objects from the burials of the Scythian elite were decorated with Greek mythological and decorative motifs.

Some of the quiver fragments preserve traces of red colour (Fig 2; 1, 2). Preliminary analyses of several fragments from Ilyinka identified the red pigment as cinnabar (HgS), a naturally occurring mercury sulphide [44]. Cinnabar was widely used by the Scythians for the decoration of a variety of different objects and materials, including wooden arrow shafts [26], boxes, and furniture, as well as bone implements such as spindles. The presence of cinnabar on the samples analysed thus may indicate the use of Scythian traditions and further points towards local production.

## Conclusion

Leather remains found in Scythian burials from 13 different 5th-4th century BCE south Ukrainian sites have been analysed for species identification. ZooMS has proven particularly useful in this investigation, providing species identifications where visual-based analysis was unsuccessful due to the removal of the pelage during leather production and the severe degradation of the artefacts.

We were able to identify a range of domesticated species (all depicted on the famous gold pectoral from Tolvsta Mohyla, Fig 4) and some wild animals. The majority of samples derive from sheep or goat, differentiable by ZooMS. Our results help us to start looking at the economics of Scythian animal exploitation–not only for food, fibre, traction, and riding, but also for the most essential but rarely considered material–leather. We also identified up to two quivers composed at least partially from human derived skin, supporting Herodotus' claim that certain parts of some Scythian quivers were actually made with human skin, perhaps from defeated foes.

## Supporting information

**S1 File.** Table S1.1. Detailed descriptions and images of samples used in the study. Table S1.2. Protein summary for sample 21 from Bulhakovo, kurgan 5, burial 2. Table S1.3. Specific peptides from sample 21 from Bulhakovo, kurgan 5, burial 2. Table S1.4. Protein summary for sample 48 from Ilyinka, kurgan 4 burial 3. Table S1.5. Specific peptides from sample 48 from Ilyinka, kurgan 4 burial 3. Table S1.6. Suspected S. vulgaris peptides in sample 48 from Ilyinka, kurgan 4 burial 3 and their Marmotini equivalents. Table S1.7. Specific peptides from sample 49 from Vil'na Ukraina 4, kurgan 22 burial 1. Table S1.8. Protein summary for sample 49 from Vil'na Ukraina 4, kurgan 22 burial 1. Text. Detailed LC-MS/MS method and discussion of results. SI references.
(PDF)

**S2 File. LC-MS/MS spectra relevant for species identification.**
(PDF)

**S1 Dataset.**
(XLSX)

## Acknowledgments

The authors thank Samantha Presslee for her help in receiving and running the MALDI plate and acknowledge the use of the Ultraflex III MALDI-ToF/ToF instrument in the York Centre of Excellence in Mass Spectrometry. Access to the material and help were generously provided by Alla Pleshivenko (Zaporizhzhia Regional Inspectorate for the Preservation of Historical and Cultural Monuments), Zoya Popandopulo and Andrei Antonov (Zaporizhzhia Regional Museum), Andrei Lopushinskij (Kherson Regional Museum), Artem Ivantsev and Irina

Koval'ova (Oles Honchar Dnipro National University), Nataliya Son and Oleksandr Shelekhan (Institute of Archaeology of the National Academy of Sciences of Ukraine). The authors thank Prof. Jesper Velgaard Olsen at the Novo Nordisk Center for Protein Research for providing access to equipment and resources.).

Some of the objects analysed in this study are stored in the museums of southern Ukraine which are currently in the war zone and at present the fate of some of them is unknown, while others are not available for further study.

## Author Contributions

**Conceptualization:** Marina Daragan, Margarita Gleba.

**Data curation:** Meaghan Mackie, Marina Daragan, Margarita Gleba.

**Formal analysis:** Luise Ørsted Brandt, Meaghan Mackie, Marina Daragan, Margarita Gleba.

**Funding acquisition:** Marina Daragan, Matthew J. Collins, Margarita Gleba.

**Investigation:** Luise Ørsted Brandt, Meaghan Mackie, Marina Daragan, Margarita Gleba.

**Methodology:** Luise Ørsted Brandt, Meaghan Mackie, Margarita Gleba.

**Project administration:** Luise Ørsted Brandt, Margarita Gleba.

**Resources:** Margarita Gleba.

**Supervision:** Margarita Gleba.

**Visualization:** Marina Daragan, Margarita Gleba.

**Writing – original draft:** Luise Ørsted Brandt, Meaghan Mackie, Marina Daragan, Margarita Gleba.

**Writing – review & editing:** Luise Ørsted Brandt, Meaghan Mackie, Matthew J. Collins, Margarita Gleba.

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
