## [Decision Letter · Decision Letter 0]

25 Aug 2023

PONE-D-23-21176Human and animal skin identified by palaeoproteomics in Scythian leather objects from UkrainePLOS ONE

Dear Dr. Brandt,

Thank you for submitting your manuscript to PLOS ONE. After careful consideration, we feel that it has merit but does not fully meet PLOS ONE’s publication criteria as it currently stands. Therefore, we invite you to submit a revised version of the manuscript that addresses the points raised during the review process.

We look forward to receiving your revised manuscript.

Kind regards,

John P. Hart, Ph.D.

Academic Editor

PLOS ONE

Journal Requirements:

2. In your manuscript, please provide additional information regarding the specimens used in your study. Ensure that you have reported human remain specimen numbers and complete repository information, including museum name and geographic location. 

For more information on PLOS ONE's requirements for paleontology and archeology research, see https://journals.plos.org/plosone/s/submission-guidelines#loc-paleontology-and-archaeology-research.

   "This work was supported by the Alexander von Humboldt Stiftung and Gerda Henkel Stiftung (project “The production technology of Scythian Archery equipment: bow, arrows and quivers”; Daragan), and the Danish National Research Foundation (grant DNRF128 PROTEIOS)."

8. We note that Figure 1 in your submission contain map/satellite images which may be copyrighted. All PLOS content is published under the Creative Commons Attribution License (CC BY 4.0), which means that the manuscript, images, and Supporting Information files will be freely available online, and any third party is permitted to access, download, copy, distribute, and use these materials in any way, even commercially, with proper attribution. For these reasons, we cannot publish previously copyrighted maps or satellite images created using proprietary data, such as Google software (Google Maps, Street View, and Earth). For more information, see our copyright guidelines: http://journals.plos.org/plosone/s/licenses-and-copyright.

Additional Editor Comments:

Both reviewers identify issues that need clarification. Please address all of their comments and suggestions while making revisions. Be sure to upload the Figure S2 file when you submit your revised manuscript.

Reviewers' comments:

Reviewer's Responses to Questions

**Comments to the Author**

1. Is the manuscript technically sound, and do the data support the conclusions?

Reviewer #1: Partly

Reviewer #2: Yes

2. Has the statistical analysis been performed appropriately and rigorously? 

Reviewer #1: N/A

Reviewer #2: Yes

3. Have the authors made all data underlying the findings in their manuscript fully available?

Reviewer #1: Yes

Reviewer #2: Yes

4. Is the manuscript presented in an intelligible fashion and written in standard English?

Reviewer #1: Yes

Reviewer #2: Yes

5. Review Comments to the Author

Reviewer #1: The manuscript “Human and animal skin identified by palaeoproteomics in Scythian leather objects from Ukraine” is an interesting application of ZooMS and LC-MS/MS for leather identification. The archaeological context, objects and interpretation of the data are well described. The results are however rushed and the paper would benefit from further details. More specifically:

There is a microscopy paragraph in Methods. The paragraph describes results rather than the method employed. Either this paragraph should be moved to results and detailed or removed entirely.

In LC-MS/MS methods, you mention searching the samples against a Swissprot database, and later that “iterative searches were made to narrow down the database to the relevant proteins from the relevant species.” This was repeated in Supplementary but no further detail is provided. The accession numbers in Supplementary for samples 48 and 49 appeared to be from NCBI. Please provide more precision on which databases were searched.

I could not find Figures S2… I only have access to S1.pdf and S1.dataset

The results paragraph is very short and some information provided in Supplementary could be moved there to offer more insights into the results.

In the species identification paragraph, it would be helpful if you could indicate the sample number when you describe specific samples, especially for the wild carnivores. I would also indicate somewhere the common name of species alongside the Latin names (in Supplementary tables as well).

It is unfortunate that only one human sample was analyzed by LC-MS/MS. The sample was chosen due to low resolution by ZooMS. It would have been interesting to compare the results on both samples by LC-MS/MS to understand why one was of lower quality. Furthermore, you have 26% of unidentified samples, which is quite high. Do you have any clues to explain the unidentified samples? Is this a problem with the method, the context in which objects were found, treatment of the leather or preservation issues?

Tables S7 and S8 have the wrong descriptions and one is missing sample name.

Table 3: “Number of proteins and corresponding proteins”. I suppose you mean corresponding peptides.

In discussion, since you mostly talk about quivers, it would be useful to have a drawing showing the different parts of the quiver. While you might not know which part of the object each sample corresponds to, as much as possible, some discussion between the species identified and corresponding part of the object would help to put the results into light. Perhaps as highlighted in a drawing/figure.

Reviewer #2: This is a very interesting study on identifying the species of origin for Scythian leather artefacts using both PMF and LC-MS-based approaches. The study is methodologically sound and the manuscript is very well written with a clear, logical flow. However, there are some minor comments that I believe the authors should consider.

---------------

Figure 2:

Do the scale bars depict cm units? Maybe add a brief sentence to the figure caption for clarification.

Results:

Inconsistency – Table 2 reads “Kurgan 4, burial 2” for sample 7 from Ilyinka which was identified as human, however the main text referring to this sample reads “kurgan 2, burial 4”. This is seen again in Figure 5 caption which also reads "kurgan 2, burial 4". Based on what the authors have presented in Table 1, should the site read "kurgan 4, burial 2" in all instances?

Supplementary Information:

Reference is made in the Methods (section Liquid chromatography-tandem mass spectrometry) and Supplementary Information S1 to Supplementary Information S2 representing spectral annotations, however S2 doesn't seem to be attached to the current manuscript submission.

Minor typographical corrections:

Orphaned round bracket “)” and extra square bracket “]” in the following Introduction sentence:

“For the purpose of the present study, Scythians are understood to be the nomads that occupied the steppes north of the Black Sea, and between the Danube and the Don Rivers, as defined by the ancient Greek ‘father of history’ Herodotus [17–19]])”

Figure 3 caption: Random comma and space after image credit.

Table 2 caption: Mustela should be italicised and capitalised as it is a genus name.

6. PLOS authors have the option to publish the peer review history of their article (what does this mean?). If published, this will include your full peer review and any attached files.

Reviewer #1: No

Reviewer #2: No

---

## [Author Response · Author response to Decision Letter 0]

8 Oct 2023

Detailed reponses to editor and reviewer comments are submitted in the uploaded Reply to Reviewers file

---

## [Decision Letter · Decision Letter 1]

25 Oct 2023

Human and animal skin identified by palaeoproteomics in Scythian leather objects from Ukraine

PONE-D-23-21176R1

Dear Dr. Gleba,

We’re pleased to inform you that your manuscript has been judged scientifically suitable for publication and will be formally accepted for publication once it meets all outstanding technical requirements.

Kind regards,

John P. Hart, Ph.D.

Academic Editor

PLOS ONE

Additional Editor Comments (optional):

Reviewers' comments:

Reviewer's Responses to Questions

**Comments to the Author**

1. If the authors have adequately addressed your comments raised in a previous round of review and you feel that this manuscript is now acceptable for publication, you may indicate that here to bypass the “Comments to the Author” section, enter your conflict of interest statement in the “Confidential to Editor” section, and submit your "Accept" recommendation.

Reviewer #1: All comments have been addressed

2. Is the manuscript technically sound, and do the data support the conclusions?

Reviewer #1: Yes

3. Has the statistical analysis been performed appropriately and rigorously? 

Reviewer #1: N/A

4. Have the authors made all data underlying the findings in their manuscript fully available?

Reviewer #1: Yes

5. Is the manuscript presented in an intelligible fashion and written in standard English?

Reviewer #1: Yes

6. Review Comments to the Author

Reviewer #1: (No Response)

7. PLOS authors have the option to publish the peer review history of their article (what does this mean?). If published, this will include your full peer review and any attached files.

Reviewer #1: No

---

## [Editor Report · Acceptance letter]

10 Nov 2023

PONE-D-23-21176R1 

Human and animal skin identified by palaeoproteomics in Scythian leather objects from Ukraine 

Dear Dr. Gleba:

I'm pleased to inform you that your manuscript has been deemed suitable for publication in PLOS ONE. Congratulations! Your manuscript is now with our production department. 

Kind regards, 

on behalf of

Dr. John P. Hart 

Academic Editor

PLOS ONE